**DOI: 10.1038/ncomms14985**　　**OPEN**

# Topological superconductivity in monolayer transition metal dichalcogenides

Yi-Ting Hsu[1], Abolhassan Vaezi[2], Mark H. Fischer[3] & Eun-Ah Kim[1]

Theoretically, it has been known that breaking spin degeneracy and effectively realizing spinless fermions is a promising path to topological superconductors. Yet, topological superconductors are rare to date. Here we propose to realize spinless fermions by splitting the spin degeneracy in momentum space. Specifically, we identify monolayer hole-doped transition metal dichalcogenide (TMD)s as candidates for topological superconductors out of such momentum-space-split spinless fermions. Although electron-doped TMDs have recently been found superconducting, the observed superconductivity is unlikely topological because of the near spin degeneracy. Meanwhile, hole-doped TMDs with momentum-space-split spinless fermions remain unexplored. Employing a renormalization group analysis, we propose that the unusual spin-valley locking in hole-doped TMDs together with repulsive interactions selectively favours two topological superconducting states: interpocket paired state with Chern number 2 and intrapocket paired state with finite pair momentum. A confirmation of our predictions will open up possibilities for manipulating topological superconductors on the device-friendly platform of monolayer TMDs.

[1] Department of Physics, Cornell University, Ithaca, New York 14853, USA. [2] Department of Physics, Stanford University, Stanford, California 94305-4060, USA. [3] Department of Condensed Matter Physics, Weizmann Institute of Science, Rehovot 7610001, Israel. Correspondence and requests for materials should be addressed to E.-A.K. (email: eun-ah.kim@cornell.edu)

The quest for material realizations of topological chiral superconductors with nontrivial Chern numbers[1–4] is fuelled by predictions of exotic signatures, such as Majorana zero modes and quantized Hall effects. Unfortunately, natural occurrence of bulk topological superconductors are rare with the best candidates being superfluid $^3$He (ref. 5) and Sr$_2$RuO$_4$ (ref. 6). Instead, much recent experimental progress relied on proximity-inducing pairing to a spin–orbit-coupled band structure building on the proposal of Fu and Kane[7]. Their key insight was that a paired state of spinless fermions is bound to be topological and that the surface states of topological insulators are spinless in that the spin degeneracy is split in position space (**r**-space): the two degenerate Dirac surface states with opposite spin textures are spatially separated. Nevertheless, despite much experimental progress along this direction[8–14], the confinement of the helical paired state to the interface of the topological insulator and a superconductor limits experimental access to its potentially exotic properties.

Another type of exotic paired states that desires material realization is the finite-momentum-paired states, which has long been pursued since the first proposals by Fulde and Ferrell[15] and by Larkin and Ovchinnikov[16]. Most efforts towards realization of such modulated superconductors[17,18], however, relied on generating finite-momentum pairing using spin-imbalance under a (effective) magnetic field in close keeping with the original proposals. Exceptions to such a spin-imbalance-based approach are refs 19,20 that made use of spinless Fermi surfaces with shifted centres. More recently, there have been proposals suggesting modulated paired states in cuprate high-$T_c$ superconductors[21–23]. However, unambiguous experimental detection of a purely modulated paired state in a solid-state system is lacking.

We note an alternative strategy that could lead to pairing possibilities for both topological and modulated superconductivity: to split the spin degeneracy of fermions in momentum space (**k**-space). This approach is essentially dual to the proposal of Fu and Kane, and it can be realized in a time-reversal-invariant non-centrosymmetric system when a pair of Fermi surfaces centred at opposite momenta $\pm \mathbf{k}_0$ consist of oppositely spin-polarized electrons (see Fig. 1a). When such a spin-valley-locked band structure is endowed with repulsive interactions, conventional pairing will be suppressed. Instead, there will be two distinct pairing possibilities: interpocket and intrapocket pairings, where the latter will be spatially modulated with pairs carrying finite centre-of-mass momentum $\pm 2\mathbf{k}_0$.

What is critical to the success of this strategy is the materialization of such **k**-space-split spinless fermions. A new opportunity has arisen with the discovery of a family of superconducting two-dimensional (2D) materials, monolayer group-VI transition metal dichalcogenides (TMDs) MX$_2$ (M = Mo, W, X = S, Se)[24–27]. Although the transition metal atom M and the chalcogen atom X form a 2D hexagonal lattice within a layer as in graphene, monolayer TMDs differ from graphene in two important ways. First, TMD monolayers are non-centrosymmetric, that is, inversion symmetry is broken (see Fig. 1b,c). As a result, monolayer TMDs are direct-gap semiconductors[28] with a type of Dresselhaus spin–orbit coupling[29,30] referred to as Ising spin–orbit coupling[31]. This spin–orbit-coupled band structure leads to the valley Hall effect[30,32], which has established TMDs as experimental platforms for pursuing valleytronics applications[30,32–36]. Our focus, however, is the fact that there is a sizable range of chemical potential in the valence band that could materialize the **k**-space spin-split band structure we desire (see Fig. 1d). Second, the carriers in TMDs have strong $d$-orbital character and, hence, correlation effects are expected to be important. Interestingly,

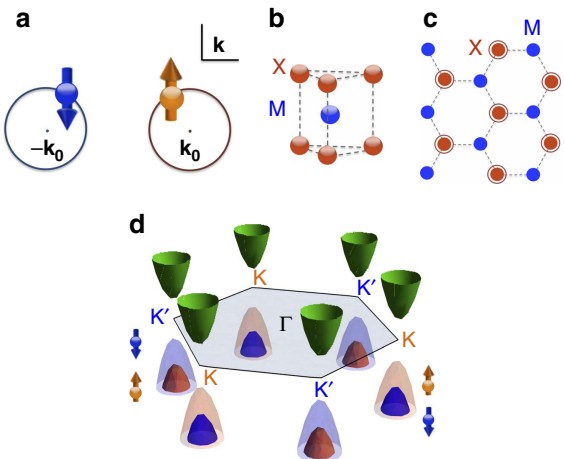

**Figure 1 | k-space spin-split in the spin-valley-locked band structure of group IV monolayer TMDs.** (**a**) Schematic Fermi surface hosting **k**-space-split spinless fermions. Here the two pockets centred at some opposite crystal momenta $\mathbf{k} = \pm \mathbf{k}_0$ host oppositely spin-polarized electrons (represented by the orange and blue arrows) in a time-reversal-symmetric manner. (**b**) A sketch for a unit cell of a monolayer TMD. The blue and red spheres represent the transition metal M atoms and the chalcogen atoms X, respectively. (**c**) A sketch for the top view of the buckled honeycomb lattice of a monolayer TMD. The blue circles represent the transition metal M atoms and the solid (hollow) red circles represent the chalcogen atoms X above (below) the plane of transition metal atoms. (**d**) Schematic low-energy dispersion of a monolayer TMD. The hexagon represents the first Brillouin zone. The green paraboloids represent the nearly spin-degenerate conduction band, and the orange and blue paraboloids represent the spin-split valence bands for the spin-up and -down electrons, respectively. This dispersion is time-reversal-symmetric since the spin-splits are opposite near the two valleys K and K′, which centred at opposite momenta $\pm \mathbf{K}$ with respect to the $\Gamma$ point.

both intrinsic and pressure-induced superconductivity have been reported in electron-doped (n-doped) TMDs[24–27] with the debate regarding the nature of the observed superconducting states still on-going[37–41].

Here we propose to obtain **k**-space-split spinless fermions by lightly hole-doping (p-doping) monolayer TMDs such that the chemical potential lies between the two spin-split valence bands. We investigate the possible paired states that can be driven by repulsive interactions[42] in such lightly p-doped TMDs using a perturbative renormalization group (RG) analysis going beyond mean-field theory[38,43]. We find two distinct topological paired states to be the dominant pairing channels: an interpocket chiral ($p$/$d$)-wave paired state with Chern number $|C| = 2$ and an intrapocket chiral $p$-wave paired state with a spatial modulation in phase. The degeneracy can be split by the trigonal warping or Zeeman effect.

## Results

**Spin-valley-locked fermions in lightly p-doped monolayer TMDs.** The generic electronic structure of group IV monolayer TMDs is shown in Fig. 1d. The system lacks inversion symmetry (see Fig. 1b,c), which leads to a gapped spectrum and a $S_z$-preserving spin–orbit coupling. Such Ising spin–orbit coupling[31] acts as opposite Zeeman fields on the two valleys that preserve time-reversal symmetry. Furthermore the spin–orbit coupling is orbital-selective[44] and selectively affects the valence band with a large spin-split[29].

By lightly p-doping the TMDs with the chemical potential $\mu$ between the spin-split valence bands, spin-valley-locked

fermions can be achieved near the two valleys (see Fig. 2a,b). Assuming negligible trigonal warping at low doping, we can use a single label $\tau = \uparrow, \downarrow$ to denote the valley and the spin. Denoting the momentum measured from appropriate valley centres $\pm \mathbf{K}$ by $\mathbf{q}$, the kinetic part of the Hamiltonian density is

$$H_0^p = \sum_{\mathbf{q},\tau} \left( -\frac{q^2}{2m} - \mu \right) c_{\mathbf{q},\tau}^\dagger c_{\mathbf{q},\tau}, \qquad (1)$$

where $\mu$ is the chemical potential, $m$ is the effective mass of the valence band and $c_{\mathbf{q},\uparrow} \equiv \psi_{\mathbf{K}+\mathbf{q},\uparrow}$ and $c_{\mathbf{q},\downarrow} \equiv \psi_{-\mathbf{K}+\mathbf{q},\downarrow}$ each annihilates a spin-up electron with momentum $\mathbf{q}$ relative to the valley centre $\mathbf{K}$ or a spin-down electron with momentum $\mathbf{q}$ relative to the valley centre $-\mathbf{K}$ (see Fig. 2a). Hence, the spin-valley-locked two-valley problem is now mapped to a problem with a single spin-degenerate Fermi pocket. Nonetheless, the possible paired states with total spin $\tau_z = \pm 1$ and $\tau_z = 0$ in fact represent the novel possibilities of intrapocket modulated pairings with total $\tau_z = \pm 1$ and interpocket pairing with total $\tau_z = 0$, respectively (see Fig. 2a,b).

**Pairing possibilities.** To discuss the pairing symmetries of the two pairing possibilities, it is convenient to define the partial-wave channels $\tilde{l}$ with respect to the two-valley centres $\pm \mathbf{K}$. Since a total spin $\tau_z = \pm 1$ intrapocket pair consists of two electrons with equal spin, Pauli principle dictates such pairing to be in a state with odd partial-wave $\tilde{l}$. Stepping back to microscopics, such pairs carry finite centre-of-mass momentum $\pm 2\mathbf{K}$ and form two copies of phase-modulated superconductor[15]. This case may or

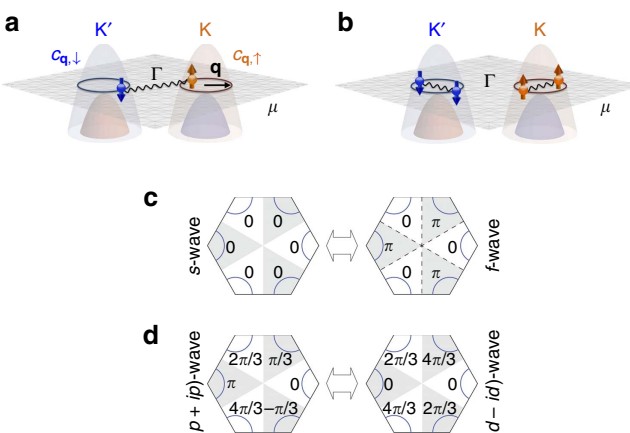

**Figure 2 | Symmetry-distinct pairing channels in a lightly p-doped monolayer TMD.** The two oppositely spin-polarized Fermi surfaces centred at K and K′ valleys (represented by the maroon and blue circles) can develop (**a**) interpocket pairing or (**b**) intrapocket pairing. Here $c_{\mathbf{q},\uparrow}$ ($c_{\mathbf{q},\downarrow}$) denotes the annihilation operator for spin-up (-down) electrons on the pocket at valley K (K′), and **q** denotes the momentum relative to the pocket centres. (**c,d**) Candidate gap functions for interpocket pairing allowed by the point group $C_{3v}$. Each hexagon represents the first Brillouin zone where the curves around the corners within the unshaded (shaded) wedges are segments of Fermi surfaces around valley K (K′). Owing to the broken $C_6$ rotations (expressed by the shaded wedges), the gap structures of (**c**) s-wave and f-wave both belong to the same irreducible representation $A_1$ and can thus mix. Similarly, the gap structures of (**d**) p-wave and d-wave both belong to the two-dimensional irreducible representation $E$ and can mix as well. The number in each wedge labels the angle corresponding to the phase of each gap function at the midpoint of the Fermi surface segment in the wedge. Note that the (p + ip)- and (d − id)-waves have the same phase-winding pattern on each pocket around respective valley centres.

may not break time-reversal symmetry due to the absence of locking between the $\tilde{l}$s of the two pockets $\tau = \uparrow, \downarrow$. For the total $\tau_z = 0$ interpocket pairing, the allowed symmetries of a superconducting state is further restricted by the underlying $C_{3v}$ symmetry of the lattice. In particular, the absence of an inversion centre allows the pairing wavefunction in each irreducible representation to be a mixture between parity-even and -odd functions with respect to the $\Gamma$ point[45]. Specifically, s-wave mixes with f-wave and d-wave mixes with p-wave (see Fig. 2c,d). Among the irreducible representations of $C_{3v}$, two fully gapped possibilities are the trivial $A_1$ representation, which amounts to (s/f)-wave pairing ($\tilde{l} = 0$) and a chiral superposition of the 2D $E$ representation, which amounts to a mixture of $p \pm ip$ and $d \mp id$ pairing ($|\tilde{l}| = 1$). The mixing implies that the non-topological f-wave channel that is typically dominant in trigonal systems as a way of avoiding repulsive interaction will be blocked together with s-wave by the repulsive interaction in the p-doped TMDs. Hence, it is clear that the pairing instability in $|\tilde{l}| = 1$ channel is all one needs for topological pairing in the p-doped TMDs.

**Two distinct topological paired states.** To investigate the effects of the repulsive interactions between transition metal d-orbitals, we take the microscopic interaction to be the Hubbard interaction, which is the most widely studied pardignamtic model of strongly correlated electronic systems

$$H'(W) = \sum_i U n_{i,\uparrow} n_{i,\downarrow}, \qquad (2)$$

where $W$ is the ultraviolet energy scale, $U > 0$ and $n_{i,s}$ is the density of electrons with spin $s$ on site $i$. By now, it is well established that the interaction that is purely repulsive at the microscopic level can be attractive in anisotropic channels for low-energy degrees of freedom, that is, fermions near Fermi surface. The perturbative RG approach has been widely used to demonstrate this principle on various correlated superconductors. For the model of p-doped TMDs defined by Equations (1) and (2), the symmetry-allowed effective interactions at an intermediate energy scale $\Lambda_0 \gtrsim 0$ close to the Fermi level in the Cooper channel (see Supplementary Note 1) would be:

$$H'_{\text{eff}}(\Lambda_0) = \sum_{\mathbf{q},\mathbf{q}',\tau,\tau'} g_{\tau,\tau'}^{(0)}(\mathbf{q}, \mathbf{q}') c_{\mathbf{q}',\tau}^\dagger c_{-\mathbf{q}',\tau'}^\dagger c_{-\mathbf{q},\tau'} c_{\mathbf{q},\tau}, \qquad (3)$$

where $\mathbf{q}$ and $\mathbf{q}'$ are the incoming and outgoing momenta. Now, the remaining task is to derive the effective inter- and intrapocket interactions $g_{\uparrow,\downarrow}(\mathbf{q},\mathbf{q}')$ and $g_{\uparrow,\uparrow}(\mathbf{q},\mathbf{q}')$ perturbatively in the microscopic repulsion $U$ and check to see whether attraction occur in the $|\tilde{l}| = 1$ channel (see Methods and Supplementary Note 2).

Before going into the details of calculation, it is important to note that isotropic pairing with $\tilde{l} = 0$ is forbidden by Pauli principle in the total $\tau_z = \pm 1$ channel and blocked by the bare repulsive interaction in the total $\tau_z = 0$ channel. Hence, we need to look for attraction in the anisotropic $\tilde{l} \neq 0$ channel, which is given by the momentum-dependent part of $g_{\tau\tau'}^{(0)}$. With our assumption of isotropic dispersion at low-doping, one needs to go to the two-loop order to find momentum dependence in the effective interaction. Fortunately, it has been known for the model of Equations (1) and (2) that effective attraction is indeed found in anisotropic channels at the two-loop order[46]. Here we carry out the calculation explicitly (see Methods and Supplementary Note 2) and find the effective interactions in the $|\tilde{l}| = 1$ channel to be attractive, that is,

$$\lambda_{\tau,\tau'}^{(0), |\tilde{l}|=1} = \frac{1}{\pi} \int_0^\pi d\theta \, g_{\tau,\tau'}^{(0)}(\theta) \Phi_1(\theta) < 0 \qquad (4)$$

for $\tau, \tau' = \uparrow, \downarrow$, where $\theta \equiv 2\sin^{-1}(\frac{|\mathbf{q} \pm \mathbf{q}'|}{2q_F})$ is the angle associated with the momentum transfer, and $\Phi_1(\theta) = \sqrt{2}\cos(\theta)$ is the normalized angular-momentum-one eigenstate in 2D.

In the low-energy limit, the effective attractions in the $|\tilde{l}| = 1$ channel at the intermediate energy scale $\Lambda_0$ in Equation (4) will lead to the following two degenerate topological paired states (see Methods): the interpocket $(p/d)$-wave pairing, which is expected to be chiral (see Fig. 3a) and the modulated intrapocket pairing (see Fig. 3b). The degeneracy is expected for the model of Equations (1) and (2) with its rotational symmetry in the pseudo spin $\tau$. There are two ways this degeneracy can be lifted. First, the trigonal warping will suppress intrapocket pairing as the two points on the same pocket with opposing momenta will not be both on the Fermi surface any more (see Fig. 3c). On the other hand, a ferromagnetic substrate will introduce an imbalance between the two pockets, which promotes intrapocket pairing[47] (see Fig. 3d).

## Discussion

The distinct topological properties of the two predicted exotic superconducting states lead to unusual signatures. The inter-pocket $|\tilde{l}| = 1$ paired state (see Fig. 3a) is topological with Chern number $|C| = 2$ because of the two pockets (see Methods). The Chern number dictates for two chiral edge modes, which in this case are Majorana chiral edge modes each carrying central charge $(1)/(2)$ (refs 1,48). This is in contrast to $d + id$ paired state on a single spin-degenerate pocket, which is another chiral superconducting state[49–54] with four chiral Majorana edge modes. An unambiguous signature of two Majorana edge

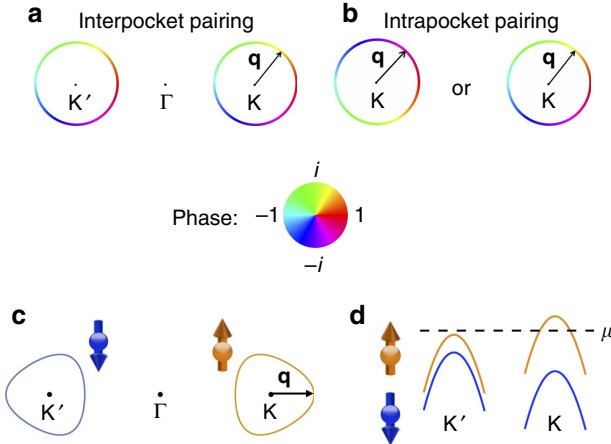

**Figure 3 | The inter- and intrapocket $|\tilde{l}| = 1$ paired states.** The gap functions of the $\tilde{l} = \pm 1$ paired states have the approximate form $q_x \pm iq_y$ on the two pockets (represented by hollow circles) centred at $\pm \mathbf{K}$, which we assume to be small and circular as discussed in the text. The colour scheme on the circles represents the phase of the gap functions, as indicated by the colour wheel. (**a**) For the interpocket pairing case, the phase winding on the two pockets are locked to each other. Overall, the paired state breaks time-reversal symmetry. (**b**) For the intrapocket pairing case, each pocket can independently have either $\tilde{l} = 1$ or $\tilde{l} = -1$, which leads to a counterclockwise or clockwise phase winding of $2\pi$. The possible factor and way to tilt the balance between the inter- and intrapocket pairings: (**c**) A sketch for the trigonally warped Fermi pockets expected upon a heavier doping where the chemical potential still lies within the spin-split. Such trigonal warping is expected to suppress the intrapocket pairing as an electron at $\mathbf{q}$ has no pairing partner on the same pocket at $-\mathbf{q}$. (**d**) The schematic low-energy dispersion near the two valleys for a monolayer TMD grown on a ferromagnetic substrate. As the chemical potential $\mu$ (represented by the dashed line) intersects only one band near one valley, the intrapocket pairing is expected to be promoted.

modes in the interpocket chiral $|\tilde{l}| = 1$ paired state will be a quantized thermal Hall conductivity[1] of

$$K_H = c \frac{\pi^2 k_B^2}{3h} T \tag{5}$$

at temperature $T$, where $c = 1$ is the total central charge. In addition, signatures of the chiral nature of such state could be revealed by a detection of time-reversal symmetry breaking in polar Kerr effect and muon spin relaxation measurements. Finally, a sharp signature of anisotropy of the pairing will be the maximization of the critical current in a direct current superconducting quantum interference device (dc SQUID) interferometry set-up of Fig. 4a at some finite flux $\Phi_{max} \neq 0$.

The intrapocket $|\tilde{l}| = 1$ paired state (see Fig. 3b) is not only topological, but also its phase of the gap is spatially modulated with $e^{i2\mathbf{K} \cdot \mathbf{r}}$ and $e^{-i2\mathbf{K} \cdot \mathbf{r}}$ for spin-up and -down pairs, respectively, where $\mathbf{r}$ is the spatial coordinate of the centre-of-mass of the pair (see Supplementary Note 3). Since the gaps on the two pockets are not tied to each other in principle, the system may be either helical respecting time-reversal symmetry ($C = 0$) or chiral ($C = 2$). Either way, there will be a Majorana zero mode of each spin species at a vortex core so long as $\tau_z$ is preserved. What makes the intrapocket paired state distinct from existing candidate materials for topological superconductivity, however, is its spatial modulation. Smoking gun signature of the modulation in phase would be the halved period $(hc)/(4e)$ of the oscillating voltage across the dc SQUID set-up in Fig. 4b in flux $\Phi$ due to the difference between the pair momenta on the two sides of the junction. Another signature of the intrapocket paired state will be the spatial profile of the modulated phase directly detected with an atomic resolution scanning Josephson tunnelling microscopy[23,55].

In summary, we propose the $\mathbf{k}$-space spin splitting as a new strategy for topological superconductivity. Specifically, we predict lightly p-doped monolayer TMDs with their spin-valley-locked band structure and correlations to exhibit topological super-conductivity. Of the monolayer TMDs, $WSe_2$ may be the most promising as its large spin-splitting energy scale[56] allows for

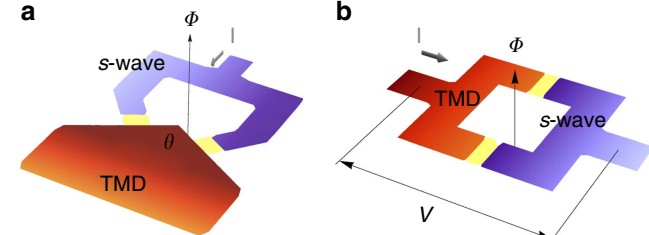

**Figure 4 | Configurations of possible SQUID experiments for probing the two paired states.** In both **a,b**, the red and blue parts indicate the lightly p-doped monolayer TMD and a uniform s-wave superconductor, respectively, which are connected by two Josephson junctions represented by the yellow strips. $I$ is the applied current and $\Phi$ is the magnetic flux through the loop. (**a**) The proposed dc SQUID interferometer set-up that can detect the anisotropy of the interpocket pairing symmetry. The flux dependence of the critical current is expected to be insensitive and sensitive to the angle $\theta$ between the edges connected to the two junctions for isotropic and anisotropic pairing, respectively. (**b**) The proposed dc SQUID interferometer set-up that can probe the finite pair momentum of the intrapocket pairs for the $C = 0$ case. The TMD is oriented in the direction such that the phase of the pairing wavefunction is spatially modulated along the junction. The period in flux $\Phi$ of the modulated voltage $V$ across the SQUID loop is expected to be halved into $(hc)/(4e)$ since the difference between the pair momenta on the two sides of a junction requires simultaneous tunnelling of a spin-up and a spin-down intrapocket pair, each carrying pair momentum $2\mathbf{K}$ and $-2\mathbf{K}$, into the uniform superconductor.

substantial carrier density within the spin-valley-locked range of doping[57]. The rationale for the proposed route is to use a lower symmetry to restrict the pairing channel. The merit of this approach is clear when we contrast the proposed setting to the situation of typical spin-degenerate trigonal systems. With a higher symmetry, trigonal systems typically deals with the need for anisotropic pairing due to the repulsive interaction by turning to the topologically trivial $f$-wave channel[2,49]. The n-doped TMDs whose low-energy band structure is approximately spin-degenerate fall into this category. Hence, experimentally realized superconductivity in n-doped systems would likely be topologically trivial even if the superconductivity is driven by the same repulsive interaction we consider here. The predicted topological paired states in p-doped TMDs are a direct consequence of the spin-valley locking, which breaks the spin degeneracy in $\mathbf{k}$-space and creates two species of spinless fermions. Experimental confirmation of the predicted topological superconductivity in p-doped TMDs will open unprecedented opportunities in these highly tunable systems.

## Methods

**Perturbative RG calculation.** For the RG calculation, we follow the perturbative two-step RG procedure in ref. 49, which has been used to study superconductivity in systems such as $Sr_2RuO_4$ (ref. 58) and generic hexagonal lattices with spin degeneracy[49]. Taking the Hubbard on-site repulsion in Equation (2) as the microscopic interaction, the first step is to integrate out higher-energy modes and obtain $g_{\tau,\tau'}^{(0)}$ in Equation (3), the low-energy effective interactions in the Cooper channel at an intermediate energy $\Lambda_0 \gtrsim 0$ close to the Fermi level. The second step is to study the evolution of these effective interactions as the energy flows from $\Lambda_0$ to 0, which is governed by the RG equations.

In the first step, we calculate the inter- and intrapocket effective interactions $g_{inter}^{(0)}(\mathbf{q},\mathbf{q}') \equiv g_{\tau,\bar{\tau}}^{(0)}(\mathbf{q},\mathbf{q}')$ and $g_{intra}^{(0)}(\mathbf{q},\mathbf{q}') \equiv g_{\tau,\tau}^{(0)}(\mathbf{q},\mathbf{q}')$ in terms of the incoming and outgoing momenta $\mathbf{q}$ and $\mathbf{q}'$ order by order in $U$ until we obtain attraction in one of them in certain partial-wave channel $\tilde{l}$. Following ref. 46, we find the effective interactions to be (see Supplementary Note 2)

$$g_{inter}^{(0)}(\mathbf{q},\mathbf{q}') \sim C + \frac{m^2 U^3}{2\pi^3}\frac{\sqrt{4q_F^2 - p'^2}}{2q_F} - \frac{U^3 m^2}{64\pi^3}\left(1 - \frac{p^2}{4q_F^2}\right)\log\left[1 - \frac{p^2}{4q_F^2}\right], \quad (6)$$

and

$$g_{intra}^{(0)}(\mathbf{q},\mathbf{q}') \sim C' - \frac{m^2 U^3}{2\pi^3}\frac{\sqrt{4q_F^2 - p^2}}{2q_F} - \frac{U^3 m^2}{64\pi^3}\left(1 - \frac{p^2}{4q_F^2}\right)\log\left[1 - \frac{p^2}{4q_F^2}\right], \quad (7)$$

where $\mathbf{p} = \mathbf{q} \pm \mathbf{q}'$ is the momentum transfer, $C > 0$ and $C' < 0$ are momentum-independent constants coming from tree level and one-loop order, and the momentum-dependent terms come solely from two-loop order.

Each partial-wave $\tilde{l}$ component is given by the projection of $g_{inter/intra}^{(0)}(\mathbf{q},\mathbf{q}')$ on to the normalized angular momentum $\tilde{l}$ eigenstate in 2D, $\Phi_{\tilde{l}}(\theta) = \sqrt{2}\cos(\tilde{l}\theta)$, where $\theta \equiv 2\sin^{-1}\left(\frac{p}{2q_F}\right)$ is the angle associated with the momentum transfer $\mathbf{p}$. We find

$$\lambda_{inter/intra}^{(0),\tilde{l}} = \frac{1}{\pi}\int_0^\pi d\theta g_{inter/intra}^{(0)}(\theta)\Phi_{\tilde{l}}(\theta)$$
$$= \frac{2\sqrt{2}\alpha}{\pi}\frac{(\pm 1)^{\tilde{l}+1}}{1 - 4\tilde{l}^2} - \frac{\beta}{\sqrt{2}\pi}\frac{H_{1-\tilde{l}} + H_{1+\tilde{l}} + 2\log 2 - 3}{\tilde{l}\left(1 - \tilde{l}^2\right)}\sin(\tilde{l}\pi), \quad (8)$$

where $H_n$ is the $n$th harmonic number and $\alpha \equiv (U^3 m^2)/(2\pi^3)$ and $\beta \equiv (U^3 m^2)/(64\pi^3)$ are positive constants related to density of states and interaction strength. Here terms with $\alpha$ and $\beta$ come from contributions with one particle–particle and one particle–hole bubble, and two particle–hole bubbles, respectively (see Supplementary Note 2). The $\alpha$ term in $\lambda_{intra}^{(0),\tilde{l}}$ acquires an extra minus sign on top of $(-1)^{\tilde{l}}$ from the closed fermion loops in Supplementary Fig.1 (3g) and (3h). Meanwhile, the $\alpha$ term in $\lambda_{inter}^{(0),\tilde{l}}$ contains an implicit $(-1)^{\tilde{l}}$ factor because of the fact that the outgoing external momenta in Supplementary Fig. 1 (3a) and (3b) are exchanged, which is equivalent to setting $\Phi_{\tilde{l}}(\theta) \to \Phi_{\tilde{l}}(\pi - \theta)$.

Note that $\lambda_{intra}^{(0),\tilde{l}}$ with even $\tilde{l}$s is forbidden since intrapocket pairs have equal spin, and that $\lambda_{intra}^{(0),\tilde{l}} = \lambda_{intra}^{(0),\tilde{l}}$ for odd $\tilde{l}$s since they correspond to the spin-triplet states with $\tau_z = 0$ and $\pm 1$, respectively. While $\lambda_{inter}^{(0),0} > 0$ as expected from the bare repulsion, the most negative values are $\lambda_{inter}^{(0),\pm 1} = \lambda_{intra}^{(0),\pm 1} \sim -0.3\alpha - 0.04\beta < 0$.

In the second step, we derive and solve the RG equations to study the evolutions of the effective interactions $\lambda_{inter/intra}^{\tilde{l}}(E)$ as the energy $E$ lowers from $\Lambda_0$ to 0. Using $\lambda_{inter/intra}^{(0),\tilde{l}}$ in Equation (8) as the initial values for the RG flows, the channel with the

most relevant attraction in the low-energy limit $E \to 0$ is the dominant pairing channel. Under the assumption that the energy contours for $0 < E < \Lambda_0$ are isotropic, different partial-wave components do not mix while the inter- and intrapocket interactions with the same $\tilde{l}$ can in principle mix. By a procedure similar to that in refs 50,59, we find the RG equations up to one-loop order to be

$$\frac{d\lambda_{inter}^{\tilde{l}}}{dy} = -(1 - d_2)\left(\lambda_{inter}^{\tilde{l}}\right)^2 \quad (9)$$

and

$$\frac{d\lambda_{intra}^{\tilde{l}}}{dy} = -(d_1 - d_3)(\lambda_{intra}^{\tilde{l}})^2 - 2d_3(\lambda_{inter}^{\tilde{l}})^2, \quad (10)$$

where the inverse energy scale $y \equiv \Pi_{pp}^{ss}(0) \sim v_0 \log(\Lambda_0/E)$ is the RG running parameter, $d_1(y) \equiv (\partial \Pi_{pp}^{ss}(\pm 2\mathbf{K}))/(\partial y)$, $d_2(y) \equiv (\partial \prod_{ph}^{ss}(\pm 2\mathbf{K}))/(\partial y)$, and $d_3(y) \equiv (\partial \Pi_{ph}^{ss}(0))/(\partial y)$. Here $\Pi_{pp/ph}^{ss'}(\mathbf{k})$ is the non-interacting static susceptibility at momentum $\mathbf{k}$ in the particle–particle or particle–hole channel defined in Supplementary Note 1. Since the low-energy band structure is well nested at $\pm 2\mathbf{K}$ in the particle–particle channel, the Cooper logarithmic divergence appears not only at $\mathbf{k} = 0$ but also $\pm 2\mathbf{K}$ (see Supplementary Note 1). Thus, $d_1(y) = 1$. On the other hand, since the low-energy band structure is poorly nested at any $\mathbf{k}$ in the particle–hole channel and is far from van Hove singularity, the particle–hole susceptibilities do not diverge in the low-energy limit (see Supplementary Note 1). Thus, $d_2(y), d_3(y) \ll 1$ in the low-energy limit $y \to \infty$. Therefore, with logarithmic accuracy, the inter- and intrapocket interactions renormalize independently with the well-known RG equation in the Cooper channel

$$\frac{d\lambda_i^{\tilde{l}}}{dy} = -\left(\lambda_i^{\tilde{l}}\right)^2 \quad (11)$$

with $i =$ inter, intra. The RG flow $\lambda_i^{\tilde{l}}(y) = \frac{\lambda_i^{(0),\tilde{l}}}{1 + \lambda_i^{(0),\tilde{l}}y}$, which solves the RG equation, shows that the pairing interaction in channel $\tilde{l}$ becomes a marginally relevant attraction only if the initial value $\lambda_i^{(0),\tilde{l}} < 0$. Since we concluded that the most negative initial values occur in the $|\tilde{l}| = 1$ channels for both inter- and intrapocket interactions in the first step of the RG procedure, we expect degenerate inter- and intrapocket $|\tilde{l}| = 1$ pairings in the low-energy limit.

**The Chern number of interpocket paired state.** The interpocket chiral $|\tilde{l}| = 1$ paired state becomes just a spinful $p + ip$ paired state with total spin $\tau_z = 0$ when we map the spin-valley-locked two-pocket problem to a spin-degenerate single-pocket problem. The spinful $p + ip$ pairing comprises two copies of 'spinless' $p + ip$ pairings as the Bogoliubov-de Gennes Hamiltonian of the former can be written as

$$H = \sum_\mathbf{q} \varepsilon_\mathbf{q}\left(c_{\mathbf{q},\uparrow}^\dagger c_{\mathbf{q},\uparrow} + c_{\mathbf{q},\downarrow}^\dagger c_{\mathbf{q},\downarrow}\right) + \Delta_\mathbf{q}\left(c_{\mathbf{q},\uparrow}^\dagger c_{-\mathbf{q},\downarrow}^\dagger + c_{\mathbf{q},\downarrow}^\dagger c_{-\mathbf{q},\uparrow}^\dagger\right) + \text{H.c.}$$
$$= \sum_\mathbf{q}\left(\varepsilon_\mathbf{q} c_{\mathbf{q},+}^\dagger c_{\mathbf{q},+} + \Delta_\mathbf{q} c_{\mathbf{q},+}^\dagger c_{-\mathbf{q},+}^\dagger + \text{H.c.}\right)$$
$$+ \left(\varepsilon_\mathbf{q} c_{\mathbf{q},-}^\dagger c_{\mathbf{q},-} - \Delta_\mathbf{q} c_{\mathbf{q},-}^\dagger c_{-\mathbf{q},-}^\dagger + \text{H.c.}\right),$$

where the low-energy dispersion $\varepsilon_\mathbf{q} = -(q^2)/(2m) - \mu$, the gap function $\Delta_\mathbf{q} \sim q_x + iq_y$ and $c_{\mathbf{q},\pm} = (c_{\mathbf{q},\uparrow} \pm c_{\mathbf{q},\downarrow})/\sqrt{2}$. Since a spinless $p + ip$ paired state has Chern number $C = 1$, where $C = \frac{1}{8\pi}\int d^2q\hat{\mathbf{m}} \cdot \left[\partial_{\mathbf{q}_x}\hat{\mathbf{m}} \times \partial_{\mathbf{q}_y}\hat{\mathbf{m}}\right]$ with $\hat{\mathbf{m}} = (\text{Re}[\Delta_\mathbf{q}], \text{Im}[\Delta_\mathbf{q}], \varepsilon_\mathbf{q})/\sqrt{\varepsilon_\mathbf{q}^2 + |\Delta_\mathbf{q}|^2}$, the $\tau_z = 0$ spinful $p + ip$ paired state in the single-pocket system has $C = 2$. Hence, the interpocket chiral $|\tilde{l}| = 1$ pairing in the two-pocket system has $C = 2$ as well.

**Data availability.** The authors declare that the data supporting the findings of this study are available within the paper and its Supplementary Information file.

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

## Acknowledgements

We thank Reza Asgari, Debdeep Jena, Katja Nowak, Grace Xing, K. T. Law and Andrey Chubukov for helpful discussions. Y.-T.H. and E.-A.K. were supported by the Cornell Center for Materials Research with funding from the NSF MRSEC programme (DMR-1120296). E.-A.K. was supported in part by by the National Science Foundation (Platform for the Accelerated Realization, Analysis, and Discovery of Interface Materials (PARADIM)) under Cooperative Agreement No. DMR-1539918. A.V. was supported by Gordon and Betty Moore Foundation and in part by Bethe postdoctoral fellowship. M.H.F. acknowledges support from the Swiss Society of Friends of the Weizmann Institute.

## Author contributions

Y.-T.H. carried out the RG calculations to identify the dominant paired states. Y.-T.H. and A.V. analysed the topological properties for the paired states. Y.-T.H. and M.H.F. analysed the pairing symmetries for the paired states. E.-A.K. supervised the project and wrote the paper with contributions from Y.-T.H., A.V. and M.H.F.

## Additional information

**Competing interests:** The authors declare no competing financial interests.

