## [Peer Review File · Nature Communications]

Reviewers' comments:

Reviewer #1 (Remarks to the Author):

In this manuscript, the authors discussed possible topological superconducting states in p-doped transition metal dichalcogenide (TMD)s from repulsive interaction between electrons. The results is obtained in a perturbative RG approach by considering up to two-loop diagrams. In my opinion, the impact of such a complicated and subtle calculation and its validity for making experimental predictions is not clear, nor is the pairing symmetry fully clarified in this study. Therefore I do not recommend its publication on Nature Communication, for reasons detailed below.

1. As mentioned in abstract, "superconductivity in electron-doped (n-type) TMDs is by now well established". Is the superconducting mechanism for n-type TMD established as the on-site Hubbard-type repulsive interaction? If yes, how is it established theoretically and confirmed experimentally? Before these questions are addressed, why should we consider superconductivity (SC) driven by Hubbard interaction in possible p-type SC in future experiments?
2. What are the symmetries in model (1)? E.g. how is time reversal symmetry and lattice rotational symmetry implemented?
3. Before the RG analysis is carried out, it helps a lot to list all possible pairing symmetries in the TMD in terms of their symmetry-breaking patterns. E.g. rotational symmetry can be spontaneously broken as in the case of $Cu_xBi_2Se_3$.
4. I find it hard to follow the arguments for suppression of s-wave pairing at the bottom of page 5.
5. The $|||=1$ channel inter- and intra-pocket pairings are energetically degenerate in the RG results. Is there any generic reason for this degeneracy? Is it due to a microscopic symmetry or emergent symmetry in the low-energy Hamiltonian? If such an emergent symmetry is non-existent in the microscopic Hamiltonian, which pairing symmetry will be favored in the end?
6. In the vortex core of intra-pocket $|||=1$ pairing state, there will be a pair of Majorana zero modes. Since the paired state generally breaks time reversal and crystal symmetries (and there is no spin rotational symmetry to start with due to SOC), there is no reason to exclude mixing of the two Majorana modes, which will gap the pair of Majorana bound states at vortex core similar to a d+id SC. So why should "k-space spin splitting protect Majorana zero modes of each spin species at a vortex core"?
7. Can the authors elucidate why trigonal warping favor inter-pocket pairing, and how can we increase trigonal warping in TMD?
8. In the perturbative RG analysis, what is the electron self-energy from the Hubbard-type interaction, and how does it affect the pairing states?
9. In appendix I.F. it was stated that "the FSs are poorly nested". This is far from obvious in all figures. Can this be justified by explicitly computing the particle-hole channel susceptibility?

Reviewer #2 (Remarks to the Author):

Dear Editor,

The study of topological superconductivity in monolayer transition metal dichalcogenides (TMD) is a timely topic. In recent experiments, superconductivity has been observed in n-doped monolayer MoS_2 , $MoSe_2$ and WS_2 . Superconductivity has also been observed in heavily p-doped monolayer $NbSe_2$ in which the chemical potential cuts through both spin-up and spin-down bands near each K point. In this manuscript, the authors explore the regime where the system is lightly p-doped in which the chemical potential cuts through only one spin split band near each K point. This is the regime which is

less explored both experimentally and theoretically.

Focusing on this lightly p-doped regime and using renormalisation group methods, the authors pointed out that repulsive interactions can induce attractive interaction in the Cooper channel. Importantly, the $l=1$ channel in the partial wave expansion dominates. This results in two possible topological superconducting phases. One is the inter-pocket $p+d$ wave pairing with Chern number equals to two. Another is the intra-pocket $p+d$ wave pairing which is a superconducting state with pair density wave.

The results found are interesting. The methods used are standard and valid. I support the publication of this work at Nature Communications.

I just have a few comments:

1. For the discussion about the time-reversal invariant intra-pocket pairing state, the authors mentioned that the Chern number = 0 phase is topological because of spin conservation in the z -direction. As a result, the two Majorana modes in the vortex states do not couple to each other and the system remains topological. I do not agree with this argument. In a realistic system, in order to achieve the p-doped regime, strong gating on the monolayer TMD is needed. As shown in recent experiments, this gating can induce Rashba type spin orbit coupling (SOC). The Rashba SOC can mix spin up and spin down states and couple to two Majorana fermions. Therefore, I suspect that the Chern number = 0 phase is not topological in the presence of Rashba SOC. I think the authors should be careful with the claim that the Chern number = 0 phase is topological.
2. The attractive interaction in the $l=1$ channel can result in the intra-pocket or inter-pocket pairing. But which pairing states have the lower energy? The discussion in the summary part is too brief to the readers.
3. There are some minor spelling mistakes in the manuscript and I believe these spelling mistakes will be corrected in later versions of the manuscript.

REVIEWERS' COMMENTS:

Reviewer #1 (Remarks to the Author):

The authors have properly addressed the questions and improved presentations in the revised manuscript, and I recommend the publication of the current version on Nature Communication.

Reviewer #2 (Remarks to the Author):

I went through the manuscript and the reply of the authors again. I still believe that the study of p-doped topological superconductivity in TMD is a very timely and important topic. The RG study of the authors goes beyond mean field theory, which is valid only for attractive interactions. Importantly, the electron-electron correlation effects in TMD can be quite significant. In TMDs with 2H structure, such as NbSe₂ and TaSe₂, superconductivity coexists with CDW. It means that interaction effects can indeed be important. Previous mean field studies, such as the one presented in Ref.38, ignored these correlation effects.

In other TMDs with different structure, such as 1T-TaS₂, the correlation effects are so strong that they can be Mott insulators [doi:10.1038/ncomms10956]. Therefore, the current theoretical study is indeed experimentally relevant.

Referee 1 raised many valid questions but I believe the authors answered his/her questions in satisfactory ways. For example, RG is probably the best analytical tool to study strongly interacting two-dimensional systems with repulsive interactions. The authors used the right tools for the study of superconducting TMD.

I can also comment on the pairing symmetry of the system. As pointed out by the authors, the $|l|=1$ pairing belongs to the E' representation of the point group D_{3h} , which is a two-dimensional representation. This is very similar to the E_u representation of the D_{3d} point group in Cu_xBi₂Se₃ pointed out by Referee 1. Due to the two-dimensional representation of the point group, one may find time-reversal symmetry breaking phases or rotational symmetry breaking phases. I believe the authors found the time-reversal symmetry breaking phase with finite Chern number which is similar to the so-called chiral phase in Cu doped Bi₂Se₃. The nematic phase, which breaks rotational symmetry, is not found by the authors. I think the authors' results are interesting from the symmetry point of view as well. The finite momentum pairing phase found by the authors has no counterparts in Cu doped Bi₂Se₃.

I recommend the publication of this work at Nature Communications.

Referee #1 (Remarks to the Author):.

“In this manuscript, the authors discussed possible topological superconducting states in p-doped transition metal dichalcogenide (TMD)s from repulsive interaction between electrons. The results is obtained in a perturbative RG approach by considering up to two-loop diagrams. In my opinion, the impact of such a complicated and subtle calculation and its validity for making experimental predictions is not clear, nor is the pairing symmetry fully clarified in this study. Therefore I do not recommend its publication on Nature Communication, for reasons detailed below.

As for the impact of the RG calculation we would like to draw the referee’s attention to the “track record” of the RG approach. First, the same type of RG calculation on heavily doped graphene [Nandkishore et al, *Nature Physics* **8** 158-163, cited 288 times] strongly motivated experimental pursuit for example, by Geim’s group [Chapman et al, *Scientific Reports* **6**, 23254]. Secondly, this type of approach and related RG approach has been the main thrust for predicting exotic s[±] pairing in Fe-based superconductors. In the superconductivity research community, it is the most respected approach.

As for the pairing symmetry, we admit that our initial discussion regarding the two degenerate channels (intra-pocket, v.s. inter-pocket) was terse. Prompted by the referee’s constructive criticism, we have expanded the discussion as we discuss below.

1. As mentioned in abstract, “superconductivity in electron-doped (n-type) TMDs is by now well established”. Is the superconducting mechanism for n-type TMD established as the on-site Hubbard-type repulsive interaction? If yes, how is it established theoretically and confirmed experimentally? Before these questions are addressed, why should we consider superconductivity (SC) driven by Hubbard interaction in possible p-type SC in future experiments? ”

Strictly speaking, the existence of superconductivity in n-doped TMDs is well established experimentally, but the pairing mechanism has not been revealed though Roldan *et al* [37] emphasized the role of repulsive interaction based on estimations from DFT calculations. At this point, one could either continue to dig deeper into the superconductivity in n-doped TMD or explore the new possibility in p-doped TMD. Our strategic reason for proposing to switch attention to p-doped TMD is the fact that the symmetry constraints in the n-doped TMD and p-doped TMD are totally different due to the band selective spin-orbit coupling. This is because the n-doped TMD is spin-degenerate whereas p-doped TMD can realize

Cornell University Department of Physics

spin-valley locking. Spin-degeneracy in the n-doped TMD puts the system in the same category as other trigonal systems, such as $K_2Cr_3As_3$ [Phys. Rev. X. **5**, 011013, arxiv:1502.03928], even when repulsive interaction is taken into account. It is well-known that such systems accommodate repulsive interaction by pairing in the f-wave channel [Phys. Rev. B. **88**, 054515, Phys. Rev. Lett. **113**, 097001, Phys. Rev. B. **81**, 224505], which is topologically trivial. On the other hand, spin-valley locking in the p-doped TMD and the resulting lower symmetry really restricts the pairing channel. Hence our prediction for topological superconductivity in the p-doped TMD is standing on a robust footing of symmetry principles. Since our predictions are guided by robust principles and the possibility is far more tantalizing than what is expected in the n-doped TMD, we firmly believe resolving p-doped TMD should not be a prerequisite for venturing into n-doped TMD.

“2. What are the symmetries in model (1)? E.g. how is time reversal symmetry and lattice rotational symmetry implemented?”

Model (1) as we presented in the initial submission is invariant under both time-reversal operation (T) and lattice three-fold rotation (C_3) as required by the point group D_{3h} of TMDs [31,38]. Recall the model $H_0(\vec{q}) = at(\tau q_x \sigma_x + q_y \sigma_y) + \frac{A}{2} \sigma_z - \lambda \tau s_z \otimes \frac{\sigma_z - \mathbb{1}_{2 \times 2}}{2}$. The Pauli matrices s_i and τ_i act on the bases of spin and atomic orbitals respectively. In such bases, T and C_3 act as $T: i\hat{s}_y \otimes \mathbb{1}_{2 \times 2}, i \rightarrow -i, \tau \rightarrow -\tau, \vec{q} \rightarrow -\vec{q}$ and $C_3: \mathbb{1}_{2 \times 2} \otimes \begin{pmatrix} 1 & 0 \\ 0 & e^{i2\pi/3} \end{pmatrix}, \tau \rightarrow \tau, q_x \rightarrow \cos \frac{2\pi}{3} q_x - \sin \frac{2\pi}{3} q_y, q_y \rightarrow \sin \frac{2\pi}{3} q_x + \cos \frac{2\pi}{3} q_y$. Under T , one can check that $T^{-1}H_0T = H_0$. Similarly, $C_3^{-1}H_0C_3 = H_0$. Thus, H_0 is time-reversal and C_3 -symmetric.

However, in order to avoid confusion we have now further simplified the discussion of the kinetic energy part of the model.

“3. Before the RG analysis is carried out, it helps a lot to list all possible pairing symmetries in the TMD in terms of their symmetry-breaking patterns. E.g. rotational symmetry can be spontaneously broken as in the case of $CuxBi_2Se_3$.”

The possible pairing symmetries are given by the irreducible representations of point group D_{3h} . The energetically more favorable (fully gapped) possible pairing symmetries belong to trivial representation A_1' ($|\tilde{l}| = 0$) and the two-dimensional representation E' ($|\tilde{l}| = 1$). Prompted by the referee's advice we have clarified the symmetry based discussion in the main text and moved technicality of the RG calculation to the supplementary. If further lowering of the symmetry is to occur, it would be in picking one element of the two-dimensional representation E' . This would require additional tendency for such symmetry breaking which is outside the scope of our model.

“4. I find it hard to follow the arguments for suppression of s-wave pairing at the bottom of page 5.”

Cornell University Department of Physics

s-wave pairing is blocked in both channels but with slightly different reasons. For intra-pocket pairing of equal spin electrons, the pairing must be in an odd- \tilde{l} channel because of the Pauli principle because a Cooper pair wave function is a two-electron wave function. Consider $\Psi(1,2) = g(\vec{q}_1, \vec{q}_2)\chi(s_1, s_2)$ where $\chi(s_1, s_2) = |\uparrow\uparrow\rangle$ is the spin part and $g(\vec{q}_1, \vec{q}_2)$ is the orbital part with $\vec{q}_1 = \vec{q}$ and $\vec{q}_2 = -\vec{q}$. $\Psi(1,2)$ needs to obey fermionic statistics. But $|\uparrow\uparrow\rangle$ is clearly symmetric under the exchange of the two electrons $1 \leftrightarrow 2$. Hence the orbital part should to be antisymmetric under $\vec{q} \leftrightarrow -\vec{q}$. For inter-pocket pairing between two opposite spins, s-wave is blocked when we are considering a purely electronic mechanism driven by repulsive interaction. This is the reason why other superconductors with carriers from transition metal ions are found and understood to be non-s-wave. We have clarified this point in the revised manuscript in line 124: “Before going into the details of calculation, it is important to note that isotropic pairing with $\tilde{l}=0$ is forbidden by Pauli principle in the total $\tau_z=\pm 1$ channel and blocked by the bare repulsive interaction in the total $\tau_z=0$ channel.”

“5. The $|\tilde{l}|=1$ channel inter- and intra-pocket pairings are energetically degenerate in the RG results. Is there any generic reason for this degeneracy? Is it due to a microscopic symmetry or emergent symmetry in the low-energy Hamiltonian? If such an emergent symmetry is non-existent in the microscopic Hamiltonian, which pairing symmetry will be favored in the end?”

These are excellent questions. The degeneracy is due to the symmetry in the dispersion centered at two valleys when trigonal warping is ignored at low doping. The new notation of pseudo-spin makes this clearer. Due to the assumed degeneracy, the Hamiltonian Eq. (1) and (3) are SU(2) symmetric under rotation of the pseudospin. Hence a degeneracy between pairings in the total $\tau_z=0$ channel and the total $\tau_z=\pm 1$ channel is expected. We now discuss two ways in which this balance can be tipped. First, trigonal warping will favor inter-pocket pairing. Second, introducing imbalance between two spin-species for instance through ferromagnetic substrate will favor intra-pocket pairing. We have now added this information with new figures Fig. 3(c) and (d) in the revised manuscript.

“6. In the vortex core of intra-pocket $|\tilde{l}|=1$ pairing state, there will be a pair of Majorana zero modes. Since the paired state generally breaks time reversal and crystal symmetries (and there is no spin rotational symmetry to start with due to SOC), there is no reason to exclude mixing of the two Majorana modes, which will gap the pair of Majorana bound states at vortex core similar to a d+id SC. So why should “k-space spin splitting protect Majorana zero modes of each spin species at a vortex core?”

The Ising-type SOC in TMDs preserves spin τ_z since the Ising SOC acts as opposite effective magnetic fields on the two pockets K and K'. Thus the two zero modes, which have up spin and down spin respectively, will not mix so long as τ_z remains a good quantum number. We have added a disclaimer in line 163: “there will be a Majorana zero mode of each spin species at a vortex core so long as τ_z is preserved” to clarify this point.

“7. Can the authors elucidate why trigonal warping favor inter-pocket pairing, and how can we increase trigonal warping in TMD?”

Cornell University Department of Physics

For intra-pocket pairing, a hole on the Fermi surface at $K+q$ should find a partner to pair at $K-q$. With trigonal warping, the dispersion is no longer rotationally symmetric around the K point. Most importantly, when $K+q$ lies on the Fermi surface, $K-q$ would be off Fermi surface and hence cannot pair with the hole at $K+q$. Nevertheless, this hole at $K+q$ can still pair with a hole at $-K-q$ [see Fig. 3(c)]. Therefore, the trigonal warping favors inter-pocket pairing. Trigonal warping is expected to increase with doping.

8. In the perturbative RG analysis, what is the electron self-energy from the Hubbard-type interaction, and how does it affect the pairing states?

We presume that the referee is asking about the normal self-energy $\Sigma^N(\vec{q})$, not the anomalous self-energy that is proportional to the superconducting gap. For the normal self-energy $\Sigma^N(\vec{q})$, the relevant term is real and only leads to a correction to the chemical potential. The irrelevant terms only modify the bare electron mass into effective mass but do not destabilize the fixed point (Fermi liquid). The Fermi liquid fixed point is only unstable against the marginally relevant pairing instability, which is given by the anomalous self-energy $\Sigma^A(\vec{q})$, which is of our interest. In other words, normal self-energy does not change the physics at discussion in any qualitative manner.

9. In appendix I.F. it was stated that “the FSs are poorly nested”. This is far from obvious in all figures. Can this be justified by explicitly computing the particle-hole channel susceptibility?

To be precise, what we meant was that the FSs in p-doped TMDs are poorly nested in the *particle-hole channel* such that particle-hole instabilities (e.g. spin density waves) are subdominant. The nesting condition in the particle-hole channel at some momentum \vec{Q} is $\epsilon_{\vec{k}} = -\epsilon_{\vec{k}+\vec{Q}}$ where $\epsilon_{\vec{k}}$ is the low energy band structure. This readily occurs in the one-dimensional system with $Q=2k_F$. However, in a multi-band two-dimensional system it requires a hole pocket and an electron pocket to have exactly flipped dispersion. In p-doped TMD's, since both pockets are hole pockets, they do not nest in the particle-hole channel. By contrast, the nesting condition for the particle-particle channel is $\epsilon_{\vec{k}} = \epsilon_{\vec{k}+\vec{Q}}$, which is perfectly satisfied (in the absence of trigonal warping). Hence the finite momentum intra-pocket pairing is the dominant finite Q instability.

Referee #2 (Remarks to the Author):

“The study of topological superconductivity in monolayer transition metal dichalcogenides (TMD) is a timely topic. In recent experiments, superconductivity has been observed in n-doped monolayer MoS₂, MoSe₂ and WS₂. Superconductivity has also been observed in heavily p-doped monolayer NbSe₂ in which the chemical potential cuts through both spin-up and spin-down bands near each K point. In this manuscript, the authors explore the regime where the system is lightly p-doped in which the chemical potential cuts through only one spin split band near each K point. This is the regime which is less explored both experimentally and theoretically.

Cornell University Department of Physics

Focusing on this lightly p-doped regime and using renormalisation group methods, the authors pointed out that repulsive interactions can induce attractive interaction in the Cooper channel. Importantly, the $l=1$ channel in the partial wave expansion dominates. This results in two possible topological superconducting phases. One is the inter-pocket p+d wave pairing with Chern number equals to two. Another is the intra-pocket p+d wave pairing which is a superconducting state with pair density wave.

The results found are interesting. The methods used are standard and valid. I support the publication of this work at Nature Communications.

Thank you for your expert opinion and interest.

I just have a few comments:

1. For the discussion about the time-reversal invariant intra-pocket pairing state, the authors mentioned that the Chern number = 0 phase is topological because of spin conservation in the z-direction. As a result, the two Majorana modes in the vortex states do not couple to each other and the system remains topological. I do not agree with this argument. In a realistic system, in order to achieve the p-doped regime, strong gating on the monolayer TMD is needed. As shown in recent experiments, this gating can induce Rashba type spin orbit coupling (SOC). The Rashba SOC can mix spin up and spin down states and couple to two Majorana fermions. Therefore, I suspect that the Chern number = 0 phase is not topological in the presence of Rashba SOC. I think the authors should be careful with the claim that the Chern number = 0 phase is topological.”

Indeed the Rashba SOC will spoil the protection. We added a disclaimer in line 163 “there will be a Majorana zero mode of each spin species at a vortex core so long as τ_z is preserved”.

“2. The attractive interaction in the $l=1$ channel can result in the intra-pocket or inter-pocket pairing. But which pairing states have the lower energy? The discussion in the summary part is too brief to the readers.”

We thank the referee for pointing this out. We now clearly point out in line 138 that the degeneracy is the result of the assumed valley-independent quadratic dispersion. Trigonal warping [see Fig. 3(c)] will disfavor the intra-pocket pairing while ferromagnetic substrate can promote intra-pocket pairing [see Fig. 3(d)].

“3. There are some minor spelling mistakes in the manuscript and I believe these spelling mistakes will be corrected in later versions of the manuscript.”

Reviewer #1 (Remarks to the Author):

The authors have properly addressed the questions and improved presentations in the revised manuscript, and I recommend the publication of the current version on Nature Communication.

We thank the first referee for his/her support.

Reviewer #2 (Remarks to the Author):

I went through the manuscript and the reply of the authors again. I still believe that the study of p-doped topological superconductivity in TMD is a very timely and important topic. The RG study of the authors goes beyond mean field theory, which is valid only for attractive interactions. Importantly, the electron-electron correlation effects in TMD can be quite significant. In TMDs with 2H structure, such as NbSe₂ and TaSe₂, superconductivity coexists with CDW. It means that interaction effects can indeed be important. Previous mean field studies, such as the one presented in Ref.38, ignored these correlation effects.

In other TMDs with different structure, such as 1T-TaS₂, the correlation effects are so strong that they can be Mott insulators [doi:10.1038/ncomms10956]. Therefore, the current theoretical study is indeed experimentally relevant.

Referee 1 raised many valid questions but I believe the authors answered his/her questions in satisfactory ways. For example, RG is probably the best analytical tool to study strongly interacting two-dimensional systems with repulsive interactions. The authors used the right tools for the study of superconducting TMD.

I can also comment on the pairing symmetry of the system. As pointed out by the authors, the $|l|=1$ pairing belongs to the E' representation of the point group D_{3h} , which is a two-dimensional representation. This is very similar to the E_u representation of the D_{3d} point group in $Cu_xBi_2Se_3$ pointed out by Referee 1. Due to the two-dimensional representation of the point group, one may find time-reversal symmetry breaking phases or rotational symmetry breaking phases. I believe the authors found the time-reversal symmetry breaking phase with finite Chern number which is similar to the so-called chiral phase in Cu doped Bi_2Se_3 . The nematic phase, which breaks rotational symmetry, is not found by the authors. I think the authors' results are interesting from the symmetry point of view as well. The finite momentum pairing phase found by the authors has no counterparts in Cu doped Bi_2Se_3 .

I recommend the publication of this work at Nature Communications.

We thank the second referee for his/her support and the expert opinion and interest.